# The Combination of Loss of ALDH1L1 Function and Phenformin Treatment Decreases Tumor Growth in *KRAS*-Driven Lung Cancer

**DOI:** 10.3390/cancers12061382

**Published:** 2020-05-28

**Authors:** Seon-Hyeong Lee, Yoon Jeon, Joon Hee Kang, Hyonchol Jang, Ho Lee, Soo-Youl Kim

**Affiliations:** 1Division of Cancer Biology, Research Institute, National Cancer Center, Goyang 10408, Korea; shlee1987@gmail.com (S.-H.L.); imyoon81@ncc.re.kr (Y.J.); wnsl2820@gmail.com (J.H.K.); hjang@ncc.re.kr (H.J.); 2Graduate School of Cancer Science and Policy, National Cancer Center, Goyang 10408, Korea

**Keywords:** aldehyde dehydrogenase 1L1, NSCLC, KRAS, cancer metabolism

## Abstract

Lung adenocarcinoma cells express high levels of *ALDH1L1*, an enzyme of the one-carbon pathway that catalyzes the conversion of 10-formyltetrahydrofolate into tetrahydrofolate and NAD(P)H. In this study, we evaluated the potential of *ALDH1L1* as a therapeutic target by deleting the *Aldh1l1* gene in *Kras^LA2^* mice, a model of spontaneous non-small cell lung cancer (NSCLC). Reporter assays revealed KRAS-mediated upregulation of the *ALDH1L1* promoter in human NSCLC cells. *Aldh1l1^−/−^* mice exhibited a normal phenotype, with a 10% decrease in *Kras*-driven lung tumorigenesis. By contrast, the inhibition of oxidative phosphorylation inhibition using phenformin in *Aldh1l1*^−/−^; *Kras^LA2^* mice dramatically decreased the number of tumor nodules and tumor area by up to 50%. Furthermore, combined treatment with pan-ALDH inhibitor and phenformin showed a decreased number and area of lung tumors by 70% in the *Kras^LA2^* lung cancer model. Consistent with this, previous work showed that the combination of *ALDH1L1* knockdown and phenformin treatment decreased ATP production by as much as 70% in NSCLS cell lines. Taken together, these results suggest that the combined inhibition of ALDH activity and oxidative phosphorylation represents a promising therapeutic strategy for NSCLC.

## 1. Introduction

Although *KRAS* is the gene most frequently mutated in lung adenocarcinoma, an effective KRAS-targeted therapy has not yet been developed [1,2]. Developing drugs to target KRAS mutations was known as “undruggable”. For years, alternative approaches have developed targeted agents affecting the signaling cascades downstream of RAS, such as MAPK and PI3K pathways. BRAF inhibitors (vemurafenib and dabrafenib) [3] and dual specificity MEK1/MEK2 inhibitors (trametinib and combimetinib) [3] have been approved as single agents. Recent promising advances in targeting KRAS G12C, such as AMG510 [4], have increased hope for approval in clinical trials as a clinical agent.

Oncogenic *KRAS* promotes cellular survival, proliferation, migration, autophagy, anabolic metabolism, and changes in the microenvironment [5,6]. The metabolic flux into the non-oxidative pentose phosphate pathway increases nucleic acid biosynthesis and activates hexosamine biosynthesis and the glycolytic pathway [7]. Bioinformatics analysis of metabolic enzymes in non-small cell lung cancer (NSCLC) revealed upregulation of aldehyde dehydrogenase (ALDH) isoforms including *ALDH1L1* [8,9]. Analysis of *ALDH1L1* expression by immunohistochemical staining showed that NSCLC cancer patients showed a higher expression level than normal control showed [10]. In folate metabolism, *ALDH1L1* (10-formyltetrahydrofolate dehydrogenase; EC 1.5.1.6), one of the most abundant folate-binding proteins [11], plays a role in converting 10-formyl-tetrahydrofolate (THF) to THF and CO_2_ with production of NAD(P)H, resulting in reduced purine synthesis in normal cells [12]. However, when mitochondrial folate metabolism is dysregulated, the cytosolic folate pathway in which *ALDH1L1* participates can support purine synthesis and proliferation in the opposite manner [12]. Formation of THF through oxidation of 10-formyl-THF, catalyzed by *ALDH1L1*, contributes to recycling of THF for purine synthesis [13]. Folate metabolism is an important metabolic pathway that produces one-carbon units for nucleic acid synthesis [14]. In addition, ~50% of NAD(P)H production in cancer cells depends on the 10-formyl-THF-pathway [15]. Hence, we investigated whether KRAS induces *ALDH1L1* to promote tumor growth. Subsequently, we assessed whether *ALDH1L1* could have the potential to be a therapeutic target by analyzing the effect of *Aldh1l1* deletion in *Kras^LA2^* mice, a surrogate model of human NSCLC.

## 2. Results

### 2.1. Aldh1l1 Expression is Associated with KRAS Mutation in Lung Cancer Cell Lines

We examined the *KRAS*-mutant NSCLC cell lines A549 (G12S), H460 (Q61H), H23 (G12C), and HOP-62 (G12C), as well as the *KRAS* wild-type (WT) lines H1299 and H1975. To confirm that *KRAS* mutation was correlated with *ALDH1L1* expression, we treated *KRAS*-mutant NSCLC cells with *KRAS* siRNA (Figure 1a). The silencing of *KRAS* reduced *ALDH1L1* levels by 70%, 70%, 60%, and 70% in H23, HOP-62, H490, and A549, respectively. Next, we confirmed that *ALDH1L1* expression was increased by the overexpression of mutant *KRAS* (Figure 1b). When the *KRAS* WT line H1299 was transfected with WT or mutant (G12D) *KRAS*, the level of *ALDH1L1* increased 2.6-fold and 2.0-fold when mutant and WT *KRAS* were over-expressed, respectively (Figure 1b left). A similar pattern was observed in H1975 cells (3.2-fold and 2.3-fold, respectively; Figure 1b right). This result suggests that the regulation of *ALDH1L1* expression in NSCLC cells depends on mutant or WT *KRAS* expression level.

### 2.2. KRAS Response Region in the ALDH1L1 Promoter

To determine whether *ALDH1L1* transcription is regulated by KRAS, we constructed a reporter vector for measuring *ALDH1L1* promoter activity using the dual-luciferase system (Figure 2a). A549, H23, H460, and HOP-62 cells stably expressing the *ALDH1L1* promoter reporter were transfected with *KRAS* siRNA. *ALDH1L1* promoter activity was decreased ~35% by *KRAS* knock-down (Figure 2b), whereas when *KRAS* was over-expressed, *ALDH1L1* promoter activity increased ~1.15- and 1.21-fold in H1299 and H1975, respectively (Figure 2c). An increase in *ALDH1L1* expression as the result of KRAS wild type overexpression is not very relevant because KRAS is not transcription factor that direct affect gene expression. However, KRAS triggers signaling cascades including Raf-MEK-Erk signaling for Myc and Max transcription factor activation or Rho-Rac-JNK for Elk1 transcription factor activation [16]. Therefore, the knockdown of KRAS showed stronger effect of *ALDH1L1* expression compared to over expression of KRAS in cancer cells, because all transcription factors, as well as downstream signaling molecules, may be required to increase the transcription activity of *ALDH1L1*.

Next, we performed an in silico analysis to predict which transcription factors acted downstream of KRAS to induce expression of the *ALDH1L1*. Previous studies reported that KRAS activates ELK1, ETS1, SP1, SP3, and RREB1 [17]. Based on our analysis, we identified possible SP1 and ELK1 binding sites in the *ALDH1L1* promoter region (Figure 2d), suggesting that the regulation of *ALDH1L1* by KRAS is mediated by activation of SP1 and ELK1. *ALDH1L1* expression was highly increased in tumor nodules of the lung tissue from the *Kras^LA2^* murine mouse model (Figure 2e).

### 2.3. Generation of Aldh1l1-Deficient Mice

To investigate the role of *ALDH1L1* in vivo, we generated *Aldh1l1*-knockout mice. Mice with a knockout one allele (tm1a) of *Aldh1l1* was received from IMPC (International Mouse Phenotyping Consortium). The *Aldh1l1* knockout allele was generated by crossing the *Aldh1l1*^tm1a^ allele with a Cre deleter strain (zp3-Cre, Jackson laboratory strain 003651). After Cre-*loxP* recombination, exons 3 and 4 of *Aldh1l1* were deleted (Figure 3a). The mutant alleles could be transmitted from both male and female *Aldh1l1* heterozygous mice, and homozygous mice could be obtained by interbreeding the heterozygotes. *Aldh1l1* heterozygous and homozygous progeny were identified by PCR genotyping (Figure 3b). As a result of the deletion of exons 3 and 4, only the first 42 of the 902 amino acids of *ALDH1L1* were correctly translated, and normal full-length *ALDH1L1* protein could not be detected in knockout mouse tissues (Figure 3c). Homozygotes did not exhibit abnormal phenotypes in most tissues, including muscle, respiratory, immune/hematopoietic, neurological/nervous, and reproductive tissues, consistent with previously reported phenotypic characterization of this mutant (https://www.mousephenotype.org/data/genes/MGI:1340024).

### 2.4. The Combination of Aldh1l1 Deficiency and Phenformin Treatment Suppresses KRAS-Driven Lung Tumorigenesis

For subsequent experiments, we used the *Kras^LA2^* murine lung cancer model, which harbors a targeted, latent *Kras^G12D^* allele that is activated by recombination [18]. The in vivo recombination event generates in an activated allele (*Kras^G12D^*) that induces varying grades of tumors, from hyperplasia to carcinomas similar to human NSCLC (Figure 4). With regard to histopathology, *Kras^LA2^* mice developed tumor progression and increased the number of nodules with age. Mice showed, predominantly, hyperplasia and adenoma, up to 32 weeks of age. As the tumor progressed, the hyperplasia ratio in nodules was decreased and the adenoma ratio in nodules was increased (Figure 4).

To explore the role of *ALDH1L1* in tumorigenesis, we genetically targeted the *Aldh1l1* gene as described above, and then crossed *Kras^LA2^* and *Aldh1l1*^−/−^ mice to generate *Kras^LA2^*; *Aldh1l1^+/+^* and *Kras^LA2^*; *Aldh1l1^−/−^* mice (Figure 5). The drug was administered once a day, 5 days a week, for 10 weeks from 6 to 16 weeks after birth of *Kras^LA2^*; *Aldh1l1^−/−^* mice. The phenotype of hyperplasia and adenoma were differentially observed between groups (Figure 5b–d). Cytokeratin 19 (CK-19) was studied as a diagnostic marker of adenocarcinoma, which is known as a marker of endometrioid adenocarcinomas, pancreatic adenocarcinoma and head and neck cancer [19] (Figure 5d). Number of nodule was decreased about 53% in *Kras^LA2^*; *Aldh1l1^−/−^* mice with phenformin treatment, while that was decreased about 9% in *Kras^LA2^*; *Aldh1l1^−/−^* mice (Figure 5e). Tumor area was also reduced about 54% in *Kras^LA2^*; *Aldh1l1^−/−^* mice with phenformin treatment while that was decreased about 14% in *Kras^LA2^*; *Aldh1l1^−/−^* mice (Figure 5f). During the experiment, we did not observe the weight loss of the mice (Figure 5g).

### 2.5. Treatment with Gossypol and Phenformin Suppresses KRAS-Driven Lung Tumorigenesis

Previously, we have shown that treatment of an NSCLC mouse xenograft model with the alcohol dehydrogenase inhibitor gossypol or mitochondrial complex I inhibitor phenformin led to only modest tumor regression, but combined treatment with both drugs led to marked synergistic tumor regression [10]. To test the therapeutic effects of gossypol and phenformin, we used *Kras^LA2^* mice, which are commonly used as a model of human NSCLC. Oral administration of vehicle or gossypol (40 mg/kg) combined with phenformin (100 mg/kg) five days per week was initiated when *Kras^LA2^* mice were at 6 weeks old and continued until the age of 16 weeks (Figure 6a). After the mice were sacrificed at the indicated times, the area and the number of tumor nodules were lower in the combined treatment group than in the control group. Sixteen-week-old *Kras^LA2^* mice exhibited hyperplasia in both the control and combined treatment groups; adenomas were detected in these animals, but adenocarcinoma lesions were not observed (Figure 6b,c). Immunohistochemical staining of cytokeratin 19 (CK-19) revealed a dramatic increase in tumor nodules, followed by tumor progression, in the *Kras^LA2^* control group (Figure 6d). Compared with *Kras^LA2^* littermates, the combined treatment group developed significantly fewer visible lung tumors from eight weeks after birth. At 16 weeks, the ratio of tumor area to normal area in the combined treatment group decreased to about one third of that the non-treated control group (Figure 6e,f), as did the number of nodules (Figure 6e,f). During the experiment, we did not observe any weight loss in the mice (Figure 6g). Together, these results demonstrate that combined treatment with gossypol and phenformin reduces lung tumor development in vivo.

Oncogenic KRAS plays a key role in controlling tumor metabolism, by changing multiple metabolic pathways to give a favor to cancer cells, including the stimulation of glucose uptake, differential channeling of glucose intermediates, reprogrammed glutamine metabolism, increased autophagy, and micropinocytosis [20]. Cells require one-carbon units for nucleotide synthesis, methylation and reductive metabolism, and these pathways support the high proliferative rate of cancer cells [21]. In this study, KRAS also induces *ALDH1L1* in the one-carbon pathway.

## 3. Discussion

In cancer cells, *KRAS* mutation induces transcription of genes encoding key enzymes of anabolic glucose metabolism, including glucose transporter 1, hexokinases, phosphofructokinase 1, and lactate dehydrogenase A [7]. *KRAS* mutation also promotes ribose biosynthesis through the non-oxidative pentose phosphate pathway (PPP), thereby decoupling ribose biogenesis without affecting cellular redox balance (NADP/NADPH ratio) [22]. One glycolytic intermediate, 3-phosphoglycerate, drives carbon into serine synthesis. Serine is required in biosynthesis of other molecules, such as glycine, cysteine, glutathione, and sphingolipids. Serine is also a major donor of one-carbon units to the folate metabolism cycle [12]. Thus, the induction of *ALDH1L1* by KRAS promotes the one-carbon pathway in NSCLC (Figure 7).

*ALDH1L1* (10-formyltetrahydrofolate dehydrogenase, EC 1.5.1.6) converts 10-formyltetrahydrofolate (10-formyl-THF) to tetrahydrofolate (THF) and CO_2_ in an NADP^+^-dependent reaction [24]. The *ALDH1L1* protein is the product of a natural fusion of three unrelated genes and consequently consists of three distinct domains: formyl dehydrogenase, 10-formyl-THF hydrolase, and ALDH [24]. Like other ALDH isotypes, *ALDH1L1* performs the aldehyde dehydrogenase reaction using NADP^+^ or NAD^+^, although the K_m_ for NAD^+^ is three orders of magnitude higher [25]. However, in the cytosol, the NADP^+^/NADPH ratio [26] is as much as three orders of magnitude lower than the NAD^+^/NADH ratio [27], because NADPH is abundantly supplied for anabolism, whereas NADH is rapidly oxidized to NAD^+^ for catabolism. Exact measurements of NADH or NADPH production by *ALDH1L1* have not been performed. NSCLC cells harboring an *ALDH1L1* knockdown produce about 10% less NADH than wild-type cells, but no change in the NADPH level was observed [10]. Treatment with the pan-ALDH inhibitor gossypol also decreases the NADH level by about 60% but does not affect the level of NADPH [10]. We observed no reduction of tumor growth in *Aldh1l1^−/−^*; *Kras^LA2^ mice*, but treatment of these mice with phenformin decreased tumor growth by ~70% (Figure 5), consistent with the 70% reduction in NADH level when *ALDH1L1*-knockdown NSCLC cells are treated with phenformin [10]. We also confirmed that combined treatment with gossypol and phenformin synergistically decreased the lung tumor area in *Kras^LA2^* mice (Figure 6). As a result of the catalytic reaction by *ALDH1L1*, NADH is yielded as a by-product from the conversion of 10-formyltetrahydrofolate to carbamate, which turns into ATP through oxidative phosphorylation [28]. We demonstrated that knock down of *ALDH1L1* using siRNA or ALDH inhibition using gossypol induced a significant reduction of ATP production in NSCLC [10]. Anti-cancer effect of gossypol alone, however, showed about 20% reduction of tumor growth, while combination treatment of gossypol and phenformin showed about 80% reduction of NSCLC tumor growth [10]. These observations are consistent with a previous report, showing that severe depletion of ATP to levels less than 25% of control triggers cell death [29].

In summary, targeting *ALDH1L1* alone did not have an anti-cancer effect in the *Kras^LA2^* lung cancer model, whereas simultaneous inhibition of *ALDH1L1* and oxidative phosphorylation significantly decreased tumor formation.

## 4. Materials and Methods

### 4.1. Cell Culture

Cell lines were obtained from the National Cancer Institute (NCI; MTA no. 2702-09). Growth medium was complete RPMI-1640, supplemented with 10% fetal bovine serum. Cells were maintained at 37 °C in a humidified incubator with 5% CO_2_.

### 4.2. Antibodies and Reagents

Anti-*ALDH1L1* (Cat. Ab56777 and Ab175198, 1:1000) and cytokerain 19 (Cat. Ab52625) were purchased from Abcam (Cambridge, UK). Anti-β-actin (Cat. Sc-47778, 1:1000) was purchased from Santa Cruz Biotechnology (Dallas, TX, USA). Anti-KRAS (Cat. H00003845-M01, 1:1000) was purchased from Abnova (Taipei, Taiwan). Control siRNA (sc-37007) and *KRAS* siRNA (Cat. Sc-35731) were purchased from Santa Cruz Biotechnology. Hs.KRAS4B (Cat. 83129) and Hs.KRAS4B G12D (Cat. 83131) were purchased from Addgene (Watertown, MA, USA). Transfection was performed with jetPEI and INTERFERin (Polyplus, New York, NY, USA). Gossypol acetic acid (Cat. G4382) and phenformin hydrochloride (Cat. P7045) were purchased from Sigma-Aldrich (St. Louis, MO, USA).

### 4.3. Immunoblotting

Whole cell lysates were prepared using RIPA buffer (Cat. R0278, Sigma-Aldrich), with protease inhibitor cocktail (P8340, Sigma-Aldrich). Protein concentrations were determined using the Bradford protein assay (Cat. 23227, Thermo Fisher Scientific, Waltham, MA, USA). Proteins were resolved by SDS-PAGE and transferred to PVDF membranes. Membranes were blocked in 5% BSA for 1 h at room temperature (RT), and then incubated overnight at 4 °C with the indicated antibodies. Membranes were washed for 1 h at RT in TBS-T, followed by incubation with a horseradish peroxidase-conjugated secondary antibody for 1 h at RT. Finally, membranes were washed for 1 h at RT in TBS-T. Protein band images were visualized using ECL reagent (Ab frontier, Seoul, Korea) on a FUSION solo (VILBER, Collégien, France).

### 4.4. ALDH1L1 Promoter Assay

The *ALDH1L1* promoter reporter construct was generated by inserting the human *ALDH1L1* promoter (including 1 kb upstream) into pLenti6-MINp-FLuc-Rluc-TKp. The pLenti6-MINp-FLuc-Rluc-TKp vector was constructed by inserting the herpes simplex virus (HSV)-thymidine kinase (TK) promoter (TKp) and *Renilla* luciferase (Rluc) in the opposite directions in the pLenti6-MINp-FLuc vector. The pLenti6-MINp-FLuc vector was constructed by inserting a minimal promoter (MINp) and firefly luciferase (Fluc) into pLenti6-GFP (Addgene Plasmid #35637) in place of the CMV enhancer, CMV promoter, and GFP.

Firefly and *Renilla* luciferase activities were measured using the Dual-Glo Luciferase Assay system E2940 (Promega, Madison, WI, USA). Briefly, after treatment with *KRAS* siRNA for 24 h, 75 μL Dual-Glo luciferase reagent was added to each well, and the plates were incubated for 10 min at room temperature. After measurement of firefly luminescence, Dual-Glo Stop & Glo reagent was added to the plate. After incubation at room temperature for 10 min, *Renilla* luminescence was measured, and the ratio of firefly to *Renilla* luminescence was calculated.

In silico transcription factor binding suite predictions were performed with ConSite (http://consite.genereg.net).

### 4.5. Spontaneous Lung Cancer Model

To assess the therapeutic effects of gossypol and phenformin, we employed the *Kras^LA2^* murine lung cancer model, which contains a targeted, latent *Kras^G12D^* allele that is activated by recombination [18]. Mice were obtained from the NCI mouse repository (strain number: 01BM3, common strain name: *Kras^LA2^*, strain nomenclature: B6.129S-*Kras*^tm3Tyj^/Nci). *Kras^LA2^* mice were backcrossed to C57BL/6 for at least six generations. After the weaning period (6 weeks after birth) the *Kras^LA2^* mice (four per group) were treated with a combination of gossypol (40 mg/kg) and phenformin (100 mg/kg) for 5 days a week, until they were sacrificed at 8, 12, and 16 weeks of age. For determination of tumor incidence and grade, whole lungs were manually inflated with 10% neutral-buffered formalin, placed in fixative for 1 day, embedded in paraffin, and sectioned. H&E staining was performed by standard procedures. Lung tumor areas were determined using ImageJ. Tumor burden was expressed as the total tumor area divided by normal lung area. This study was reviewed and approved by the Institutional Animal Care and Use Committee (IACUC) of National Cancer Center Research Institute. NCCRI is an AAALAC International accredited facility and abides by the Institute of Laboratory Animal Resources (ILAR) guide. The ethical code is NCC-15-277 (19 September 2016), NCC-17-277B (30 June 2018) and NCC-18-277B (31 August 2019).

### 4.6. Automated Immunohistochemistry

Immunohistochemistry assays were performed on a VENTANA Discovery XT automated staining instrument (Ventana Medical Systems, Tucson, AZ, USA). Slides were de-paraffinized using EZ Prep solution (Ventana Medical Systems) for 30 min at 75 °C. Epitope retrieval with CC1 solution (Ventana Medical Systems) was performed for 64 min at 95 °C. Antibodies were first titered over a range of concentrations to provide the optimum ratio of specific staining to background staining. Once titers were set, antibodies were transferred with diluent to user-fillable dispensers for use on the automated stainer. Anti-*ALDH1L1* (ab175198, 1:50) and cytokeratin 19 (ab52625, 1:1000) antibodies were acquired from Cell Signaling Technology (Danvers, MA, USA). Slides were developed using the OptiView DAB detection kit (Ventana Medical Systems). Briefly, samples were incubated with inhibitor for 8 min, linker for 8 min, multimer for 12 min, DAB/peroxide for 8 min, and copper for 4 min. The slides were then counterstained for 8 min with hematoxylin II (Ventana Medical Systems). Antibody titers were determined for each antibody using positive and negative control tissues, according to the manufacturer’s instructions.

## 5. Conclusions

KRAS upregulates the expression of *ALDH1L1* in NSCLC cells, which exhibited an increase of *ALDH1L1* in Kras-driven lung cancer model. Although targeting *ALDH1L1* alone did not have an anti-cancer effect in the *Kras^LA2^* lung cancer model, simultaneous inhibition with gossypol and phenformin significantly decreased tumor formation. These results suggest that combined inhibition of ALDH activity and oxidative phosphorylation represents a promising therapeutic strategy for NSCLC.

## Figures and Tables

**Figure 1 cancers-12-01382-f001:**
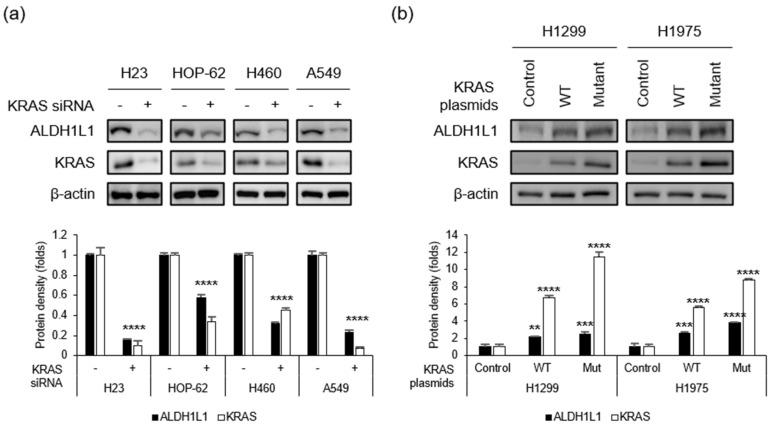
KRAS regulates *ALDH1L1* expression. (**a**) KRAS-mutant NSCLC cells were treated with control or *KRAS* siRNA, and then subjected to immunoblotting with the indicated antibodies. (**b**) *KRAS* WT NSCLC cells were transfected with control (empty vector), WT *KRAS*, or mutant (G12D) *KRAS* and subjected to immunoblotting with the indicated antibodies. Quantifications of protein density were presented as mean ± standard deviation (*n* = 3, vs. control, ** *p* < 0.01, *** *p* < 0.001 and **** *p* < 0.0001). Detailed information about western blot can be found at Appendix A.

**Figure 2 cancers-12-01382-f002:**
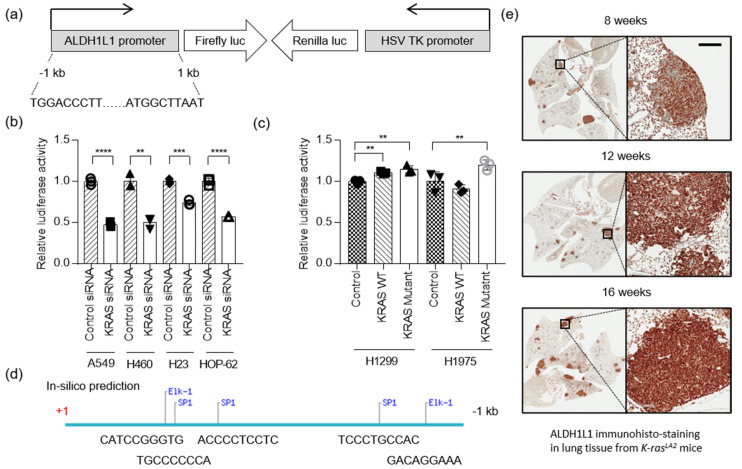
*ALDH1L1* is a target of oncogenic KRAS. (**a**) Schematic representation of the reporter construct used in the cell-based transduction system. This lentiviral reporter construct expressed firefly luciferase under the control of the *ALDH1L1* promoter and *Renilla* luciferase under the control of the HSV *TK* promoter. (**b**) *KRAS*-mutant NSCLC cells stably expressing the reporter system described in (**a**) were treated with control or *KRAS* siRNA. Firefly and *Renilla* luciferase activities were normalized against the corresponding levels in the sample transfected with control siRNA. Bars show relative luciferase activity (*n* = 3). (**c**) *KRAS* WT NSCLC cells stably expressing the reporter system described in (**a**) were transfected with control (empty vector), *KRAS* WT, or *KRAS* mutant (G12D). Firefly and *Renilla* luciferase activities were normalized against the corresponding levels in the sample transfected with empty vector. Bars indicate relative luciferase activity (*n* = 3). (**d**) In silico prediction of binding sites for transcription factors downstream of KRAS: SP1, transcription factor Sp1; Elk-1, ETS Like-1 transcription factor. (**e)** Immunohistochemical staining of *ALDH1L1* in lungs isolated from *Kras^LA2^* mice at 8, 12, and 16 weeks of age. The scale bar represents 200 mm (** *p* < 0.01, *** *p* < 0.001 and **** *p* < 0.0001).

**Figure 3 cancers-12-01382-f003:**
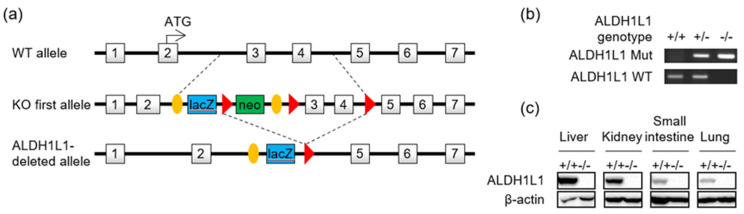
Generation of *Aldh1l1*-knockout mice. (**a**) Strategy for generating *Aldh1l1* knockout mice. (**b**) *Aldh1l1* gene disruption was confirmed by PCR genotyping. Tail genomic DNA was amplified, with specific primers for wild-type (311 bp) and mutant (307 bp) *Aldh1l1* alleles. +/+, wild-type; +/−, heterozygous; −/−, homozygous (knockout). (**c**) Immunoblotting analysis showing the absence of *ALDH1L1* in protein extracts of tissues from an *Aldh1l1^−/−^* mouse. Detailed information about western blot can be found at Appendix A.

**Figure 4 cancers-12-01382-f004:**
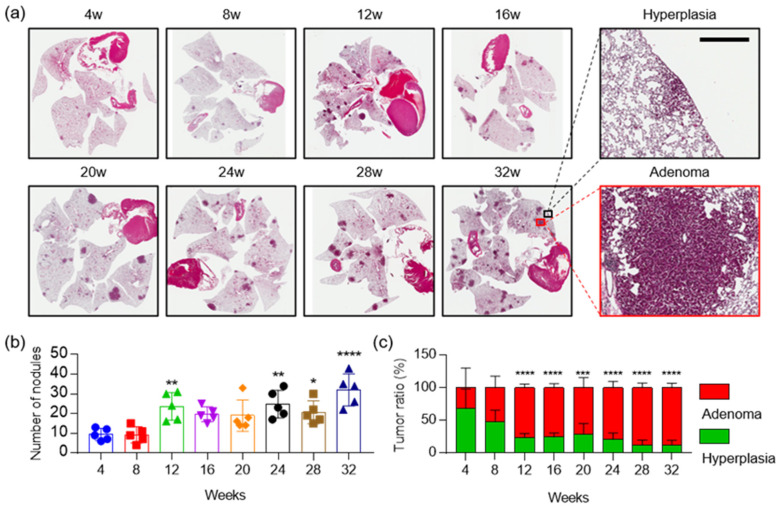
Characterizations of *Kras^LA2^* mice lungs. (**a**) Representative photomicrographs of hematoxylin and eosin (H&E) staining in mouse lungs, harvested from 4 to 32 weeks after birth. Black box indicates hyperplastic lesion and red box indicates adenoma in 32 weeks of *Kras^LA2^* mice lung, respectively. The scale bar represents 400 μm. (**b**) Quantitative analysis of number of tumor nodules in mouse lungs from 4 to 32 weeks after birth (*n* = 5). (**c**) *Kras^LA2^* mice lungs were analyzed and all lesions were classified for hyperplasia and adenoma. *p*-values were obtained by one-way or two-way ANOVA test and are indicated by asterisks (* *p* < 0.05, ** *p* < 0.01, *** *p* < 0.001 and **** *p* < 0.0001).

**Figure 5 cancers-12-01382-f005:**
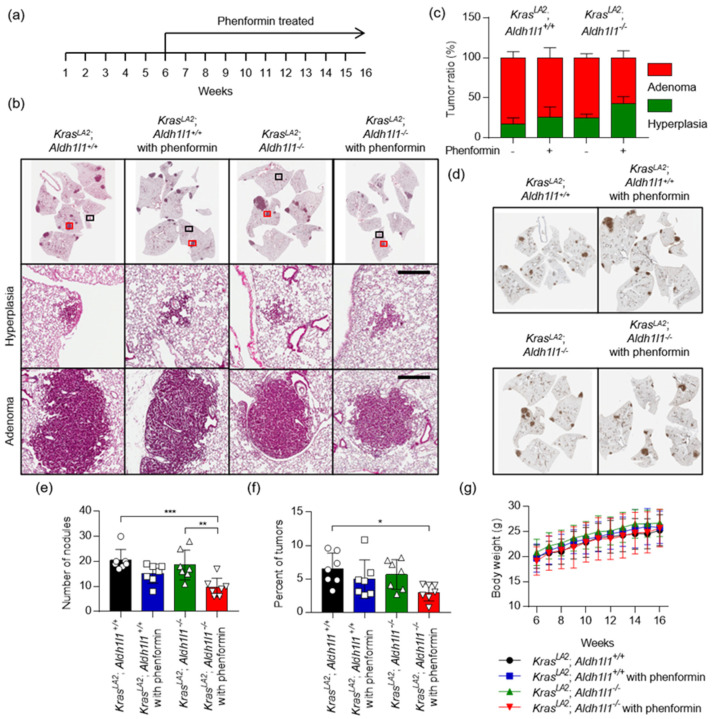
Deletion of *Aldh1l1* in combination with phenformin treatment suppresses *Kras*-driven lung tumorigenesis. (**a**) Scheme of the experimental protocol. *Kras^LA2^*; *Aldh1l1^+/+^* or *Kras^LA2^*; *Aldh1l1^−/−^* mice were treated with phenformin (100 mg/kg) or vehicle control by oral administration. Drug was administered once a day, 5 days a week, for 10 weeks from 6 to 16 weeks after birth. (**b**) Representative photomicrographs of hematoxylin and eosin (H&E) staining (Top), hyperplasia lesion (Middle, black box) and adenoma lesion (Bottom, red box) in mouse lungs harvested 16 weeks after birth. Scale bar = 400 μm. (**c**) Lungs of mice were analyzed, and all lesions were classified for hyperplasia and adenoma. **(d**) Representative photomicrographs of cytokeratin 19 staining in mouse lungs harvested 16 weeks after birth. Quantitative analysis of (**e**) number of tumor nodules and (**f**) tumor area in mouse lungs 16 weeks after birth. Tumor area (expressed as %) was calculated by dividing the total tumor area by the total area of the lung. *p*-values were obtained by one-way ANOVA test and are indicated by asterisks (* *p* < 0.05; ** *p* < 0.01; *** *p* < 0.001). (**g**) Body weight was measured once a week (*n* = 7).

**Figure 6 cancers-12-01382-f006:**
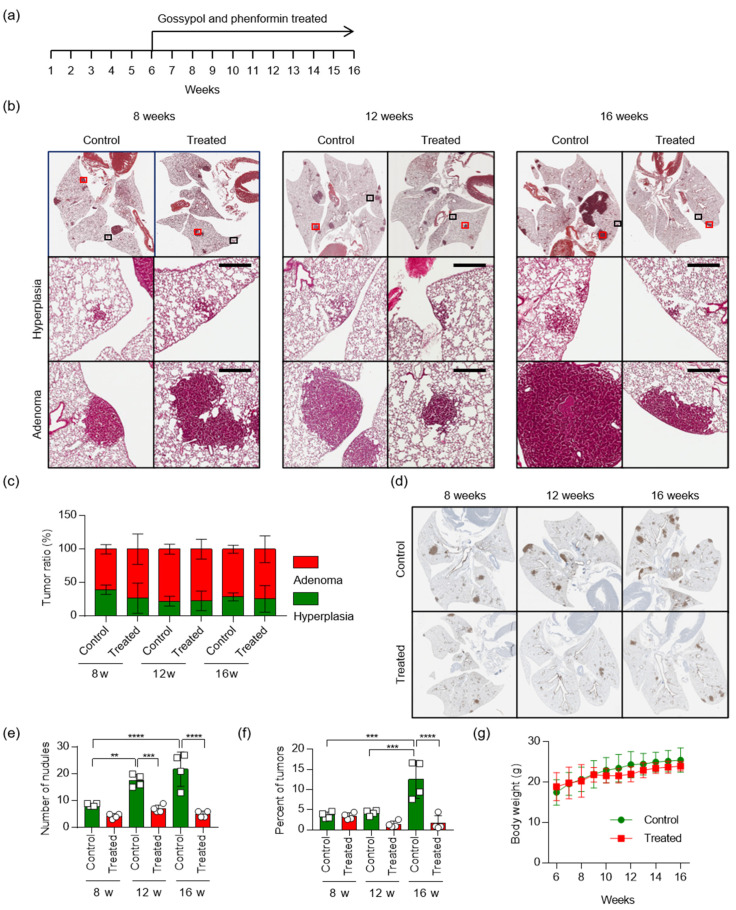
Combined treatment with gossypol and phenformin suppresses *KRAS*-driven lung tumorigenesis in mice. (**a**) Scheme of the experimental protocol. *Kras^LA2^* mice were divided into two groups: one group was the control, and the other was treated with gossypol (40 mg/kg) and phenformin (100 mg/kg) by oral administration once a day, 5 days a week, for 10 weeks, from 6 to 16 weeks after birth. (**b**) Representative photomicrographs of hematoxylin and eosin (H&E) staining (Top), hyperplasia lesion (Middle, black box) and adenoma lesion (Bottom, red box) in mouse lungs harvested 16 weeks after birth. Scale bar = 400 μm. (**c**) Lungs of mice were analyzed, and all lesions were classified for hyperplasia and adenoma. (**d**) Representative photomicrographs of cytokeratin 19 staining in mouse lungs, harvested 16 weeks after birth. (**e**) Quantitative analysis of the number of tumor nodules and (**f**) tumor area (expressed as %) in the lung of control (*n* = 4) or gossypol/phenformin treated (*n* = 4) *Kras^LA2^* mice at 8, 12, and 16 weeks after birth. Tumor burden (expressed as %) was calculated by dividing the total tumor area by the total area of the lung. *p*-values were obtained by one-way ANOVA test and are indicated by asterisks (** *p* < 0.01; *** *p* < 0.001; **** *p* < 0.0001). (**g**) Body weight was measured once a week (*n* = 4).

**Figure 7 cancers-12-01382-f007:**
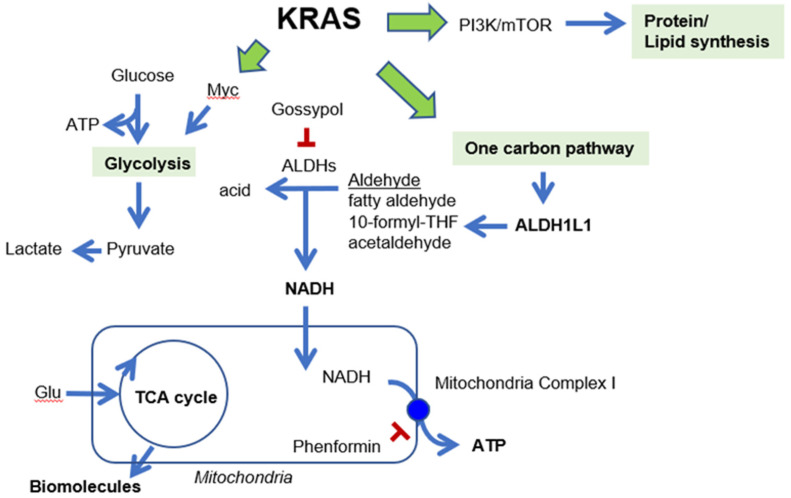
KRAS promotes glycolysis, protein and lipid synthesis and one carbon pathway [23]. Induction of *ALDH1L1* by KRAS promotes the one-carbon pathway in NSCLC. Gossypol and phenformin reduces NADH and ATP production, respectively.

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
