# Peer review of "The Combination of Loss of ALDH1L1 Function and Phenformin Treatment Decreases Tumor Growth in KRAS-Driven Lung Cancer"

_cancers, 2020, doi:10.3390/cancers12061382_

Round 1

Reviewer 1 Report

Dear Editor-in-chief,

below the comments on the manuscript (Manuscript ID: cancers-787297) by Seon-Hyeong Lee , Yoon Jeon , Joon Hee Kang , Hyonchol Jang , Ho Lee  and Soo-Youl Kim entitled "The combination of loss of ALDH1L1 function and phenformin treatment decreases tumor growth in KRAS-driven lung cancer".

In this manuscript, the authors report a potential pharmacological (gossypol and phenformin)  anti-tumoral treatment  of lung cancer (mouse model).  Particularly, they report that ALDH1L1 is a potential therapeutic target in KRAS-driven lung cancer. The study is conducted firstly in vitro cell cultures (a panel of various cell lines with KRAS different mutational background) and then in two transgenic knockout mouse model KRASLA2 and KRASLA2/Aldh1l1 -/-

They use in vitro cell lines model (siRNA and recombinant over-expression) to shown that KRAS regulates ALDH1L1 expression (Figure 1) and luciferase assay reporter to establish that ALDH1L1 is target of oncogenic KRAS (Figure 2)  (luciferase assay; in silico prediction; immunohistochemistry). Then, they move on in vivo data from two mouse knockout mice (home-made Aldh1l1 -/- knockout mice and KRASLA2 murine lung cancer model by Tyler Jacks's lab) (Figure 3, Figure 4, Figure 5).

First issue, whether specific metabolic pathways are essential for the tumorigenesis of KRAS-driven cancers are questions which remain to be answered. 

Second issue,it is not well established that ALDH1L1 is down regulated in lung cancers. The bibliography is poor and references therein report exclusively the stemness capacity associated with ALDH1L1 expression [7,8]. Otherwise, in recent literature ALDH1L1 is strongly and ubiquitously down-regulated in malignant tumors via promoter methylation. In lung tumours too. 

The topic of this manuscript is very interesting as there is considerable experimental evidence supporting a role for a mutual interplay between KRAS and ALDH1L1 in lung tumorigenesis.  Although the classical framework of the study is good, data throughout this paper are not convincing enough.

This study suffers from several draw-backs that hampered its potential significance in cancer research:

  • in vivo models are not fully characterised neither the crossed mice.The technical details should be expanded and clarified to ensure that readers understand exactly what the researchers studied.
  • pharmacological treatment with gossypol and phenformin are not described enough

Author Response

Reviewer 1

Q1. First issue, whether specific metabolic pathways are essential for the tumorigenesis of KRAS-driven cancers are questions which remain to be answered. 

We have added the following sentence in the result part p15

“Ocogenic KRAS plays a key role in controlling tumor metabolism by changing multiple metabolic pathways to give a favor to cancer cells including stimulation of glucose uptake, differential channeling of glucose intermediates, reprogrammed glutamine metabolism, increased autophagy, and micropinocytosis [1]. Oncogenic KRAS also induces nonoxidative pentose phosphate pathway (PPP) flux to fuel increased nucleic acid biosynthesis and activates the hexosamine biosynthesis and glycosylation pathways [1]. Cells require one-carbon units for nucleotide synthesis, methylation and reductive metabolism, and these pathways support the high proliferative rate of cancer cells [2]. In this study, KRAS also induces ALDH1L1 in the one-carbon pathway.”

Q2. Second issue, it is not well established that ALDH1L1 is down regulated in lung cancers. The bibliography is poor and references therein report exclusively the stemness capacity associated with ALDH1L1 expression [7,8]. Otherwise, in recent literature ALDH1L1 is strongly and ubiquitously down-regulated in malignant tumors via promoter methylation. In lung tumours too. 

A2. We have changed the sentence as “Analysis of ALDH1L1 expression by immunohistochemical staining showed that NSCLC cancer patients showed higher expression level than normal control showed [3] “ and reference with our publication “Oncotarget. 2016 Aug 2; 7(31): 49397–49410”. We have positive results with ALDH1L1 increase in lung cancer tissue in Figure 1. However, we do not have conclusion with overall survival yet.

(C) Representative immunohistochemical staining of ALDH1L1 in normal and cancerous lung tissue. Scale bar = 100 μm. Expression of ALDH1L1in cancerous (Cancer) and normal lung type I and II pneumocytes (Control). *p < 0.001, n = 57 for each case. [Oncotarget. 2016 Aug 2; 7(31): 49397–49410]

Q3. The topic of this manuscript is very interesting as there is considerable experimental evidence supporting a role for a mutual interplay between KRAS and ALDH1L1 in lung tumorigenesis.  Although the classical framework of the study is good, data throughout this paper are not convincing enough.

This study suffers from several draw-backs that hampered its potential significance in cancer research

  • in vivo models are not fully characterised neither the crossed mice. The technical details should be expanded and clarified to ensure that readers understand exactly what the researchers studied.

We have added a new Figure 4 for the explanation of time course of KRASLA2 mouse at p11 as “With regard to histopathology, KrasLA2 mice developed tumor progression and increased the number of nodules with age. Mice showed predominantly hyperplasia and adenoma up to 32 weeks of age. As the tumor progresses, hyperplasia ratio in nodules was decreased and adenoma ratio in nodules was increased (Figure 4).”

Figure 4. Characterizations of KrasLA2 mice lungs. (a) Representative photomicrographs of hematoxylin and eosin (H&E) staining in mouse lungs harvested from 4 to 32 weeks after birth. Black box indicates hyperplastic lesion and red box indicates adenoma in 32 weeks of KrasLA2 mice lung, respectively. The scale bar represents 400 μm. (b) Quantitative analysis of number of tumor nodules in mouse lungs from 4 to 32 weeks after birth (n=5). (c) KrasLA2 mice lungs were analyzed and all lesions were classified for hyperplasia and adenoma. p-values were obtained by one-way or two-way ANOVA test and are indicated by asterisks (*p,0.05, **p<0.01, ***p<0.001 and ****p<0.0001).

Figure 5 and 6 were changed and results of Figure 5 was added as

“To explore the role of ALDH1L1 in tumorigenesis, we genetically targeted the Aldh1l1 gene as described above, and then crossed KrasLA2 and Aldh1l1-/- mice to generate KrasLA2; Aldh1l1+/+ and KrasLA2;Aldh1l1-/- mice (Figure 5). Drug was administered once a day, 5 days a week, for 10 weeks from 6 to 16 weeks after birth of KRASLA2;Aldh1l1-/- mice. The phenotype of hyperplasia and adenoma were differentially observed between groups (Figure 5b-d). Cytokeratin 19 (CK-19) was studied as a diagnostic marker of adenocarcinoma which is known as a marker of endometrioid adenocarcinomas, pancreatic adenocarcinoma and head and neck cancer [4] (Figure 5d). Number of nodule was decreased about 53 % in KrasLA2;Aldh1l1-/- mice with phenformin treatment while that was decreased about 9 % in KrasLA2;Aldh1l1-/- mice (Figure 5e). Tumor area was also reduced about 54 % in KrasLA2;Aldh1l1-/- mice with phenformin treatment while that was decreased about 14 % in KrasLA2;Aldh1l1-/- mice (Figure 5f). During the experiment, we did not observe weight loss of the mice (Figure 5g).”

  • pharmacological treatment with gossypol and phenformin are not described enough

We have added a following sentence in the discussion (p20)

“As a result of the catalytic reaction by ALDH1L1, NADH is yielded as a by-product from the conversion of 10-formyltetrahydrofolate to carbamate, which turns into ATP through oxidative phosphorylation [5]. We demonstrated that knock down of ALDH1L1 using siRNA or ALDH inhibition using gossypol induced significant reduction of ATP production in NSCLC [3]. Anti-cancer effect of gossypol alone, however, showed about 20% reduction of tumor growth while combination treatment of gossypol and phenformin showed about 80% reduction of NSCLC tumor growth [3]. These observations are consistent with a previous report showing that severe depletion of ATP to levels less than 25 % of control triggers cell death [6].”

Reviewer 2 Report

The manuscript by Lee et al, showing that combined inhibition of ALDH1L1 activity and oxidative phosphorylation represents a promising therapeutic strategy for lung cancer may be an interesting issue, however it has major mistakes.

In the Introduction section authors should explain whether KRAS-targeted therapy in lung cancer has been tried and, if so, the possible problems it present.

Results 2.1. Data suggest that ALDH1L1 increase in more dependent on KRAS expresión levels than on KRAS status (mutated o wild type) (“When the KRAS WT line H1299 was 56 transfected with WT or mutant (G12D) KRAS, the level of ALDH1L1 increased 2.6-fold and 2.0-fold 57 when mutant and WT KRAS were over-expressed, respectively (Figure 1b left”)). Authors should reconsidered the formulation of this conclusion.

Results 2.2. Increase in ALDH1L1 expression as the result of KRAS wild type overexpression is not very relevant (“whereas when KRAS was over-expressed, ALDH1L1 promoter activity increased ~1.15- and 1.21-fold in H1299 and H1975, respectively”) . Authors should explain this in the manuscript.

Results 2.4. Line 115. Authors should explain the time-course appearance of lung hyperplasia or NLSCLC in KRASLA2 mice.

Results 2.5. Lines 119-121. “Treatment of an NSCLC mouse xenograft model with the alcohol dehydrogenase inhibitor gossypol or phenformin led to only modest tumor regression, but combined treatment with both drugs led to marked synergistic tumor regression” This paragraph is not understood. Authors must include a reference if they refer to an already
established experiment or remove it.

Lines 127-128 . “Sixteen-week-old KRASLA2 mice exhibited hyperplasia in both the control and combined treatment groups; adenomas were detected in these animals, but adenocarcinoma lesions were not observed (Figure 4b)”. Authors must show the histological analysis of tumors at higher magnification, together with an explanation of the tumor phenotype; tumors must be analyzed by an expert pathologist.
Lines 129-130. “Immunohistochemical staining of cytokeratin 19 (CK-19) revealed a dramatic increase in tumor nodules, followed by tumor progression, in the KRASLA2 control group

(Figure 4c).” Authors must explain whether the increase in “tumor nodules” refers to tumor burden, tumor size etc. They also should explain the significance of CK-19 immunostaining as well as include data about the antibody used.

Lines 136-137. “Together, these results demonstrate that combined treatment with gossypol and phenformin suppress lung tumor development in vivo”. Results obtained by authors show the combined treatment reduces lung tumor development, but not suppress it. Authors should changes this conclusion.

In the manuscript authors do not explain the experiments not the results corresponding to Figure 5. “Figure 5. Deletion of Aldh1l1 in combination with phenformin treatment suppresses KRAS-driven lung tumorigenesis.” Therefore authors do not explain the relevance for lung cancer development of the deletion of ALDH1L1 in KRAS mutant mice, which is the issue of the manuscript.

Author Response

Reviewer 2

Q1. In the Introduction section authors should explain whether KRAS-targeted therapy in lung cancer has been tried and, if so, the possible problems it present.

We have added the following sentence in the introduction p3.

“Developing drugs to target KRAS mutations was known as “undruggable”. For years, alternative approaches have developed targeted agents affecting the signaling cascades downstream of RAS, such as MAPK and PI3K pathways. BRAF inhibitors (vemurafenib and dabrafenib) [7] and dual specificity MEK1/MEK2 inhibitors (trametinib and combimetinib) [7] have been approved as single agents. Recent promising advances in targeting KRAS G12C such as AMG510 [8] have increased hope for approving in clinical trials as a clinical agent.”

Q2. Results 2.1. Data suggest that ALDH1L1 increase in more dependent on KRAS expresión levels than on KRAS status (mutated o wild type) (“When the KRAS WT line H1299 was 56 transfected with WT or mutant (G12D) KRAS, the level of ALDH1L1 increased 2.6-fold and 2.0-fold 57 when mutant and WT KRAS were over-expressed, respectively (Figure 1b left”)). Authors should reconsidered the formulation of this conclusion.

We have changed the conclusion as “This result suggests that regulation of ALDH1L1 expression in NSCLC cells depends on mutant or WT KRAS expression level.”

Q3. Results 2.2. Increase in ALDH1L1 expression as the result of KRAS wild type overexpression is not very relevant (“whereas when KRAS was over-expressed, ALDH1L1 promoter activity increased ~1.15- and 1.21-fold in H1299 and H1975, respectively”) . Authors should explain this in the manuscript.

We have changed the result as follows at p7.

“Increase in ALDH1L1 expression as the result of KRAS wild type overexpression is not very relevant because KRAS is not transcription factor that direct affect gene expression. However, KRAS triggers signaling cascades including Raf-MEK-Erk signaling for Myc and Max transcription factor activation or Rho-Rac-JNK for Elk1 transcription factor activation [9]. Therefore, knockdown of KRAS showed stronger effect of ALDH1L1 expression compared to over expression of KRAS in cancer cells because all transcription factors as well as downstream signaling molecules may be required to increase transcription activity of ALDH1L1.”

Q4. Results 2.4. Line 115. Authors should explain the time-course appearance of lung hyperplasia or NLSCLC in KRASLA2 mice.

We have added a new Figure 4 for the explanation of time course of KRASLA2 mouse at p11 as “With regard to histopathology, KrasLA2 mice developed tumor progression and increased the number of nodules with age. Mice showed predominantly hyperplasia and adenoma up to 32 weeks of age. As the tumor progresses, hyperplasia ratio in nodules was decreased and adenoma ratio in nodules was increased (Figure 4).”

Figure 4. Characterizations of KrasLA2 mice lungs. (a) Representative photomicrographs of hematoxylin and eosin (H&E) staining in mouse lungs harvested from 4 to 32 weeks after birth. Black box indicates hyperplastic lesion and red box indicates adenoma in 32 weeks of KrasLA2 mice lung, respectively. The scale bar represents 400 μm. (b) Quantitative analysis of number of tumor nodules in mouse lungs from 4 to 32 weeks after birth (n=5). (c) KrasLA2 mice lungs were analyzed and all lesions were classified for hyperplasia and adenoma. p-values were obtained by one-way or two-way ANOVA test and are indicated by asterisks (*p,0.05, **p<0.01, ***p<0.001 and ****p<0.0001).

Q5. Results 2.5. Lines 119-121. “Treatment of an NSCLC mouse xenograft model with the alcohol dehydrogenase inhibitor gossypol or phenformin led to only modest tumor regression, but combined treatment with both drugs led to marked synergistic tumor regression” This paragraph is not understood. Authors must include a reference if they refer to an already
established experiment or remove it.

We have changed to “Previsously we have shown that treatment of an NSCLC mouse xenograft model with the alcohol dehydrogenase inhibitor gossypol or mitochondrial complex I inhibitor phenformin led to only modest tumor regression, but combined treatment with both drugs led to marked synergistic tumor regression [3].”(p14)

Q6. Lines 127-128 . “Sixteen-week-old KRASLA2 mice exhibited hyperplasia in both the control and combined treatment groups; adenomas were detected in these animals, but adenocarcinoma lesions were not observed (Figure 4b)”. Authors must show the histological analysis of tumors at higher magnification, together with an explanation of the tumor phenotype; tumors must be analyzed by an expert pathologist.
Lines 129-130. “Immunohistochemical staining of cytokeratin 19 (CK-19) revealed a dramatic increase in tumor nodules, followed by tumor progression, in the KRASLA2 control group

Figure 5 and 6 were changed with high mag pictures.

Figure 5. Deletion of Aldh1l1 in combination with phenformin treatment suppresses KRAS-driven lung tumorigenesis. (a) Scheme of the experimental protocol. KRASLA2; Aldh1l1+/+ or KRASLA2; Aldh1l1-/- mice were treated with phenformin (100 mg/kg) or vehicle control by oral administration. Drug was administered once a day, 5 days a week, for 10 weeks from 6 to 16 weeks after birth. (b) Representative photomicrographs of hematoxylin and eosin (H&E) staining (Top), hyperplasia lesion (Middle, black box) and adenoma lesion (Bottom, red box) in mouse lungs harvested 16 weeks after birth. Scale bar = 400 μm. (c) Lungs of mice were analyzed, and all lesions were classified for hyperplasia and adenoma. (d) Representative photomicrographs of cytokeratin 19 staining in mouse lungs harvested 16 weeks after birth. Quantitative analysis of (e) number of tumor nodules and (f) tumor area in mouse lungs 16 weeks after birth. Tumor area (expressed as %) was calculated by dividing the total tumor area by the total area of the lung. p-values were obtained by one-way ANOVA test and are indicated by asterisks (*p<0.05; **p<0.01; ***p<0.001). (g) Body weight was measured once

a week (n=7).

Figure 6. Combined treatment with gossypol and phenformin suppresses KRAS-driven lung tumorigenesis in mice. (a) Scheme of the experimental protocol. KRASLA2 mice were divided into two groups: one group was the control, and the other was treated with gossypol (40 mg/kg) and phenformin (100 mg/kg) by oral administration once a day, 5 days a week, for 10 weeks from 6 to 16 weeks after birth. (b) Representative photomicrographs of hematoxylin and eosin (H&E) staining (Top), hyperplasia lesion (Middle, black box) and adenoma lesion (Bottom, red box) in mouse lungs harvested 16 weeks after birth. Scale bar = 400 μm. (c) Lungs of mice were analyzed, and all lesions were classified for hyperplasia and adenoma. (d) Representative photomicrographs of cytokeratin 19 staining in mouse lungs harvested 16 weeks after birth. (e) Quantitative analysis of the number of tumor nodules and (f) tumor area (expressed as %) in the lung of control (n=4) or gossypol/phenformin treated (n=4) KRASLA2 mice at 8, 12, and 16 weeks after birth. Tumor burden (expressed as %) was calculated by dividing the total tumor area by the total area of the lung. p-values were obtained by one-way ANOVA test and are indicated by asterisks (**p<0.01; ***p<0.001; ****p<0.0001). (g) Body weight was measured once a week (n=4).

Q7. (Figure 4c).” Authors must explain whether the increase in “tumor nodules” refers to tumor burden, tumor size etc. They also should explain the significance of CK-19 immunostaining as well as include data about the antibody used.

Following sentence is deleted because tumor burden is not relevant to number of nodules. “Tumor burden was analyzed in terms of the number of nodules and the ratio of tumor area to total normal area (Figure 5d). “

CK19 was added in the result of Figure 5 (p11-14)

“To explore the role of ALDH1L1 in tumorigenesis, we genetically targeted the Aldh1l1 gene as described above, and then crossed KrasLA2 and Aldh1l1-/- mice to generate KrasLA2; Aldh1l1+/+ and KrasLA2;Aldh1l1-/- mice (Figure 5). Drug was administered once a day, 5 days a week, for 10 weeks from 6 to 16 weeks after birth of KrasLA2;Aldh1l1-/- mice. The phenotype of hyperplasia and adenoma were differentially observed between groups (Figure 5b-d). Cytokeratin 19 (CK-19) was studied as a diagnostic marker of adenocarcinoma which is known as a marker of endometrioid adenocarcinomas, pancreatic adenocarcinoma and head and neck cancer [4] (Figure 5d). Number of nodule was decreased about 53 % in KrasLA2;Aldh1l1-/- mice with phenformin treatment while that was decreased about 9 % in KrasLA2;Aldh1l1-/- mice (Figure 5e). Tumor area was also reduced about 54 % in KrasLA2;Aldh1l1-/- mice with phenformin treatment while that was decreased about 14 % in KrasLA2;Aldh1l1-/- mice (Figure 5f). During the experiment, we did not observe weight loss of the mice (Figure 5g).”

We have added information of antibody of CK19 at 4.2. Antibodies and Reagents : cytokerain 19 (Cat. Ab52625) were purchased from Abcam (Cambridge, UK). 4.6. Automated immunohistochemistry : Anti-ALDH1L1 (ab175198, 1:50) and cytokeratin 19 (ab52625, 1:1,000) antibodies were acquired from Cell Signaling Technology (Danvers, MA, USA).

Q8. Lines 136-137. “Together, these results demonstrate that combined treatment with gossypol and phenformin suppress lung tumor development in vivo”. Results obtained by authors show the combined treatment reduces lung tumor development, but not suppress it. Authors should changes this conclusion.

We have changed to “reduces”.

Q9. In the manuscript authors do not explain the experiments not the results corresponding to Figure 5. “Figure 5. Deletion of Aldh1l1 in combination with phenformin treatment suppresses KRAS-driven lung tumorigenesis.” Therefore authors do not explain the relevance for lung cancer development of the deletion of ALDH1L1 in KRAS mutant mice, which is the issue of the manuscript.

We have added following sentence and Figure 7 at p18.

Figure 7. KRAS promotes glycolysis, protein and lipid synthesis and one carbon pathway [10]. Induction of ALDH1L1 by KRAS promotes the one-carbon pathway in NSCLC. Gossypol and phenformin reduces NADH and ATP production respectively

Reviewer 3 Report

The paper reports the effect of combination of ADLD1L1 functional impairment and the drug phenformin in decreasing tumor growth in KRAS-driven lung cancer in murine models.

The paper strengthens the role of cell metabolism in controlling tumor development and growth.

The paper is well written and results sound.

  • Is there any functional change in tumor cell behavior (cell survival, proliferation, invasiveness, gene transcription) in relation with changes in ALDH1L1 and KRAS? If the authors provide additional data on cell cultures, this will improve the force of the results.
  • Figure 1a and Figure 2 b and c need the statistical evaluation (error bars, number of replicates and asterisks for significance)
  • In Figure 4 and 5 please substitute “Schematic” (an adjective) with “Scheme” or “Schematic representation”

Author Response

Reviewer 3

  • Is there any functional change in tumor cell behavior (cell survival, proliferation, invasiveness, gene transcription) in relation with changes in ALDH1L1 and KRAS? If the authors provide additional data on cell cultures, this will improve the force of the results.

At this moment, we do not have any extra evidence of functional changes including survival, invasiveness an so on. That will be investigated in near future.

  • Figure 1a and Figure 2 b and c need the statistical evaluation (error bars, number of replicates and asterisks for significance)

We have changed the Figure1 and 2 with statistical evaluation.

  • In Figure 4 and 5 please substitute “Schematic” (an adjective) with “Scheme” or “Schematic representation”

We have changed to Scheme.

Round 2

Reviewer 1 Report

The authors have satisfactorily responded to all my questions and made the necessary changes to the manuscript.

Reviewer 2 Report

Authors have satisfactorily answered the requirements

Reviewer 3 Report

All the referees' coments have been justified or satisfied. The manuscritp has been improved.